# Elucidating ozone and PM$_{2.5}$ pollution in Fenwei Plain reveals the co-benefits of controlling precursor gas emissions in winter haze

Chunshui Lin[1], Ru-Jin Huang[1,2,3,4*], Haobin Zhong[1,5], Jing Duan[1], Zixi Wang[3,6], Wei Huang[1], and Wei Xu[1]

[1]State Key Laboratory of Loess and Quaternary Geology (SKLLQG), Center for Excellence in Quaternary Science and Global Change, Institute of Earth Environment, Chinese Academy of Sciences, Xi'an 710061, China
[2]Institute of Global Environmental Change, Xi'an Jiaotong University, Xi'an 710049, China
[3]University of Chinese Academy of Sciences, Beijing 100049, China
[4]Laoshan Laboratory, Qingdao 266061, China
[5]School of Advanced Materials Engineering, Jiaxing Nanhu University, Jiaxing 314001, China.
[6]State Key Laboratory of Atmospheric Boundary Layer Physics and Atmospheric Chemistry, Institute of Atmospheric Physics, Chinese Academy of Sciences, Beijing 100029, China

*Correspondence to*: Ru-Jin Huang (rujin.huang@ieecas.cn)

**Abstract.** Fenwei Plain, home to 50 million people in central China, is one of the most polluted regions in China. In 2018, Fenwei Plain is designated as one of the three key regions for the "Blue Sky Protection Campaign", along with the Beijing-Tianjin-Hebei (BTH) and Yangtze River Delta (YRD) regions. However, compared to BTH and YRD, our understanding of the current status of air pollution in the Fenwei Plain is limited partly due to a lack of detailed analysis of the transformation from precursor gases to secondary products including secondary organic aerosol (SOA) and ozone. Through the analysis of 7 years (2015-2021) of surface monitoring of the air pollutants in Xi'an, the largest city in the Fenwei Plain, we show that roughly 2/3 of the days exceeded either the PM$_{2.5}$ or the O$_3$ level-1 air quality standard, highlighting the severity of air pollution. Moreover, an increase in O$_3$ pollution in the winter haze was also revealed, due to the constantly elevated reactive oxygenated volatile organic compounds (OVOCs), and in particular formaldehyde with ozone formation potential of over 50 μg m$^{-3}$ in combination with the reduced NO$_2$. The abrupt decrease of NO$_2$, as observed during the lockdown in 2020, provided real-world evidence of the control measures, targeting only NO$_x$ (70% decrease on average), were insufficient to reduce ozone pollution because reactive OVOCs remained constantly high in a VOC-limited regime. Model simulation results showed that with NO$_2$ reduction from 20-70%, the self-reaction rate between peroxy radicals, a pathway for SOA formation, was intensified by up to 75%, while the self-reaction rate was only reduced with a further reduction of VOCs of >50%. Therefore, a synergic reduction in PM$_{2.5}$ and O$_3$ pollution can only be achieved through a more aggressive reduction of their precursor gases. This study elucidates the status of ozone and PM$_{2.5}$ pollution in one of the most polluted regions in China, revealing a general trend of increasing secondary pollution i.e., ozone and SOA in winter haze. Controlling precursor gas emissions is anticipated to curb both ozone and SOA formation which will benefit not just the Fenwei Plain but also other regions in China.

## 1 Introduction

The Fenwei Plain (about 760 km in length and 40–100 km in width) is the largest plain in the middle reaches of the Yellow River. It is home to over 50 million people in central China, surrounded by the Chinese Loess Plateau to the northwest and Qinling Mountains to the south. The rapid growth and urbanization of Fenwei Plain are accompanied by air pollution that is characterized by high concentrations of fine particulate matter (PM$_{2.5}$) in the heating season and high concentrations of ground-level ozone in the warm season (Elser et al., 2016; Lin et al., 2021; Song et al., 2021; Lin et al., 2022; Lin et al., 2022). Recently, air pollution in Fenwei Plain has been found to be more severe than in the Beijing-Tianjin-Heibei (BTH) region, Yangtze River Delta (YRD) region, and Pearl River Delta (PRD) region, making Fenwei Plain one of the most polluted regions in China (Cao and Cui, 2021). In addition to being the emission hotspot of air pollutants, the unique topography of the Fenwei Plain is

favorable for accumulating the air pollutants inside the basin (Cao and Cui, 2021), rapidly building up high levels of air pollutants e.g., under typical cold-haze conditions in calm weather with a wind speed of less than 2 m s$^{-1}$. In 2018, the Fenwei Plain was designated as one of the three key regions for the "Blue Sky Protection Campaign" (the other two are the BTH and YRD regions). To evaluate the effectiveness of the clean air policies and to further develop cost-effective mitigation policies in the Fenwei Plain, a better understanding of the trend in pollution patterns, sources, and formation mechanism of key pollutants i.e., PM$_{2.5}$ and ozone, is required.

As the largest city in the Fenwei Plain, Xi'an city is home to over 12 million people, with severe winter haze pollution events being recorded frequently, featuring high PM$_{2.5}$ concentration levels rivaling that in Beijing (Elser et al., 2016; Lin et al., 2021; Wang et al., 2022). However, ozone is generally viewed as a summertime problem, with very low concentrations in winter (Li et al., 2019). Most ozone studies in the Fenwei Plain were conducted in summer, with few studies performed in winter (Song et al., 2021; Yan et al., 2021; Li et al., 2022). In a recent study, Li et al. (2021) show a tendency of increasing winter-spring ozone with winter haze events in the North China Plain due to the reduced NO$_x$ emission, while the emission of reactive volatile organic compounds (VOCs) remained constantly high. In particular, the increased association of high ozone with winter haze events is highlighted during the strict lockdown period in January and February 2020, whereas the formation of secondary organic aerosol (SOA), a major fraction of PM$_{2.5}$ which is produced from the oxidation of VOCs, was found to be largely increased (He et al., 2020; Zhao et al., 2020; Duan et al., 2021; Li et al., 2021; Zhong et al., 2021). The city to nationwide lockdown provides a real-world experiment to study the impact of emission control, which mostly targeted NO$_x$ (Li et al., 2021), on the haze and ozone pollution, which, however, does not appear to be abating in the Fenwei Plain (Duan et al., 2021; Zhong et al., 2021).

Reactive VOCs and oxygenated VOCs (OVOCs) are key precursor gases of ozone and SOA (Li et al., 2022; Wang et al., 2022), the formation of which involves a series of photochemical reactions of VOC/OVOCs and NO$_x$. Briefly, upon solar radiation at wavelength <424 nm, NO$_2$ is photolyzed in the atmosphere (Li et al., 2019; Li et al., 2022). Following the photolysis of NO$_2$, the resulting O atom quickly becomes ozone, while NO$_2$ is formed from the oxidation of NO by peroxy radicals, including the peroxy radical (RO$_2$) and hydroperoxyl radical (HO$_2$), which are initiated by the oxidation of VOCs/OVOCs by the hydroxyl radical (OH) (Wang et al., 2009; Wang et al., 2019; Li et al., 2022). High levels of NO$_x$, as currently experienced in most Chinese cities, can contribute to radical termination (e.g., OH+NO$_2$) (Li et al., 2022; Wang et al., 2022), while, in NO$_x$-lean environments, RO$_2$ mainly reacts with other peroxy radicals (i.e., self-reactions), some of which can lead to SOA formation albeit not the only way (Zhao et al., 2018; Lyu et al., 2022). Previous studies in the Fenwei Plain mostly focus on hydrocarbon-like VOCs, while the OVOCs were rarely studied partly due to the limitation of sampling instrumentation (Song et al., 2021; Li et al., 2022). However, OVOCs were found to be the dominant species in other urban areas in China, showing high ozone and SOA formation potential (Li et al., 2019; Luo et al., 2020; Li et al., 2021). With the development of the state-of-the-art instrument e.g., Vocus proton-transfer-reaction time-of-flight mass spectrometer (PTR-TOF) (Krechmer et al., 2018), a better characterization of OVOCs is crucial for the co-benefits of ozone and PM$_{2.5}$ reduction in the Fenwei Plain. This is particularly important as the synergic control of ozone and PM$_{2.5}$ pollution is one of the key policies to improve air quality during the 14th 5-year Plan period (2021-2025).

In this study, a trend analysis of PM$_{2.5}$ and ozone from 2015 to 2021 were performed to understand the current status of air pollution. To disentangle the meteorological impacts on the measured trend of PM$_{2.5}$ and ozone, a machine learning (Grange et al., 2018; Dai et al., 2021; Shi et al., 2021) based meteorological normalization technique was applied, revealing the trend of de-weathered PM$_{2.5}$ and ozone. Under the background of decreasing anthropogenic emissions, changes in the chemical composition of PM$_{2.5}$ and the organic aerosol (OA) factors were investigated based on the available studies using online aerosol mass spectrometry (Elser et al., 2016; Zhong et al., 2020; Duan et al., 2021; Duan et al., 2022). As a case study, the ozone formation potential of OVOCs was analyzed based on its reactivity (Carter, 2010), while the SOA formation potential was analyzed using a 0-D chemical box modeling (Wolfe et al., 2016) in different NO$_x$ and VOC reduction scenarios. Based on the

85 ground surface and satellite observation, in combination with the modeling results, the reduction of reactive VOCs/OVOCs for the co-benefits of ozone and PM$_{2.5}$ reduction are discussed.

## 2. Method

### 2.1 Surface measurement data

Hourly O$_3$, PM$_{2.5}$, NO$_2$, SO$_2$, and CO were continuously monitored at 13 sites within the city center of Xi'an from 2015 to
90 2021 (Fig. S1). Measurements of these pollutants were routinely managed by the China National Environmental Monitoring Centre. The distance between the sampling sites is up to 40 km. Despite the distance, the time series of PM$_{2.5}$ and O$_3$ monitored at different sites were well correlated with similar magnitude in concentrations (Fig. S2). The consistency of the measurements between different sampling sites confirmed the data quality. The data from 13 sites were averaged to get the city-wide mean concentration of each pollutant and were used to perform the trend analysis.

VOC and OVOCs were measured using a Vocus-PTR which was equipped with a Long Time-of-Flight (LToF) mass analyzer, from 24 January to 6 February 2021 along with a high-resolution Long Time-of-Flight Aerosol Mass Spectrometry (AMS). The equipped LToF mass analyzer of Vocus-PTR had a mass-resolving power of approximately 12,000. Vocus-PTR measured organic vapors with a wide range of volatilities using a low-pressure reagent ion source and focusing ion-molecule reactor (FIMR). FIMR focused ions to the central axis efficiently with the quadrupole radio frequency (RF) field inside, greatly
improving the detection efficiency of product ions. Moreover, no humidity dependence for sensitivity was found for Vocus-PTR since a high-water mixing ratio was applied in the FIMR. To protect the microchannel plate detector from degrading too quickly, the big segmented quadrupole (BSQ) was set up as a high-pass band filter because the focusing effect of RF fields caused the low mass signal to be very high. More instrument details are available in Krechmer et al. (2018). In this study, Vocus-PTR was calibrated with VOC/OVOC standard mixture (Table S1). The sensitivity of Vocus-PTR towards uncalibrated
VOC compounds was calculated from the kinetic rate constant following Krechmer et al. (2018). Uncertainties for the calibrated VOC/OVOC species were 15%, while the uncertainty was 30% for the uncalibrated VOC compounds (Table S2) with known reaction rate constant ($k_{PTR}$) (https://kb.tofwerk.com/tofware/; last access: 1 March 2023). Due to the low transmission efficiency for low-mass molecules that caused high uncertainties in quantification, formaldehyde (HCHO) was scaled to the mean concentration of the surface HCHO based on satellite observation using the empirical relationship (Zhang
et al., 2012). Vocus-PTR data was only available in 2021 and was used as a case study to perform an analysis of ozone formation potential and 0-D box modeling of self-reaction rate between peroxy radicals in different scenarios.

The chemical composition of non-refractory particulate matter (NR-PM) was measured using an AMS or an aerosol chemical speciation monitor (ACSM) at the old campus of the Institute of Earth Environment, Chinese Academy of Sciences (Table S3) in the winter of 2012-2014 (Zhong et al., 2020) (Elser et al., 2016), and more recently in 2018-2019 (Duan et al., 2022), 2020
(Duan et al., 2021), and 2021 (this study; Fig. S3). For AMS measurement, Elser et al. (2016) first deployed an AMS with a PM$_{2.5}$ inlet and comprehensively evaluated its performance during the winter campaign in Xi'an. Based on the same design, a soot particle AMS (SP-AMS) equipped with a PM$_{2.5}$ inlet was recently deployed at the same sampling site, quantifying PM$_{2.5}$ chemical composition with high mass and temporal resolution (Lin et al., 2021; Duan et al., 2022). OA factors were apportioned using the Positive Matrix Factorization (PMF) with the Multilinear Engine (ME-2) (Elser et al., 2016; Zhong et
al., 2020; Duan et al., 2021; Duan et al., 2022), Similarly, OA source apportionment was performed for the 2021 dataset using the Igor Pro (WaveMetrics Inc.) - based interface of SoFi (version 8.2.1) (Canonaco et al., 2013). The apportioned factors explained the input OA matrix well (Fig. S4-S6).

### 2.2 Satellite and meteorological data

Vertical column densities (molecules cm$^{-2}$) of tropospheric formaldehyde and NO$_2$ were obtained from Sentinel-5P Level-3 Near Real-Time dataset based on the observation of the TROPOspheric Monitoring Instrument (TROPOMI) using the Google Earth Engine (Gorelick et al., 2017). This dataset was used to study the spatiotemporal variation of reactive OVOCs as a response to strict lockdown measures implemented in 2020. As a comparison, satellite images were also obtained for the same period in 2019 and 2021.

Boundary layer height (blh) and downward ultraviolet (UV) radiation at the surface was accessed from European Centre for Medium-Range Weather Forecasts (ECMWF) Reanalysis v5 (ERA5). Wind speed, wind direction, relative humidity, and air temperature were obtained from the Integrated Surface Database (ISD).

## 2.3 Meteorological normalization

A machine-learning based meteorological normalization using the random forest (RF) algorithm (Grange et al., 2018) was
used to decouple the meteorological impacts on the observed O$_3$ and PM$_{2.5}$. Firstly, the random forest was grown using the meteorological and time variables as input. The meteorological variables included the data obtained from ERA5 and ISD, while time variables included date unix, day julian, weekday, hour of the day, and day of the lunar year. 80% of the dataset was randomly selected to train the model, while 20% was used to validate the model. The "rmweather" R packages was applied during the random forest modeling (Grange et al., 2018). As shown in Fig. S7 and S8, the predicted and measured O$_3$ and
PM$_{2.5}$ were well correlated (R$^2$>0.89), suggesting the developed model rebuilt the measurements very well.

RF - based meteorological normalization was first introduced by Grange and Carslaw (2019). However, it is not straightforward to investigate the seasonal variation in the trends of de-weathered air pollutants using the proposed normalization method (Grange and Carslaw, 2019). Vu et al. (2019) enhanced the meteorological normalization procedure by repeatedly resampling the meteorological variables for a particular time point within a four-week period and the resampled variables were fed to the
developed RF model. The meteorological normalization method proposed by Vu et al. (2019), is widely applied in different sampling sites (Dai et al., 2021; Shi et al., 2021). In this study, we resampled the meteorological variables by averaging the meteorological variables for the same time point across 7 years. Our proposed method was more straightforward and subject to less variation given that meteorological variables varied a lot over a four-week period (e.g., see the comparison between the resampled and observed temperatures in Fig. S19). Also, our method is less time-consuming. Therefore, the meteorological
normalization technology assumed the meteorological variables were invariant across the years. In this scenario, the predicted values represent the changes in air pollutant concentrations that were not affected by meteorological variables (i.e., de-weathered pollutants).

## 2.4 0-D box modelling

Master Chemical Mechanism (MCM) v3.3.1 (Jenkin et al., 2003; Jenkin et al., 2019) was incorporated in a chemical box
model using the framework for 0-D Atmospheric Modeling (F0AM) version 4.2.2 (Wolfe et al., 2016). Surface observational data were averaged for the overlapping period with OVOC measurements to get the diurnal profiles of each pollutant. The averaged air pollutants, including NO$_2$, O$_3$ (not included when it was simulated, Fig. S10), CO, and the measured VOC/OVOCs (Table S2) were used to constrain the model at hourly resolution (base run). The box model reproduced the measured ozone reasonably well (Fig. S10) after optimizing the correction of the simulated photolysis rate. However, due to the uncertainties
of the box model which did not consider factors like regional transport and heterogenous chemistry, as well as the parameterized ozone deposition, the modelling results were only used to understand the atmospheric chemistry involving OVOCs by comparing different scenarios. Note that HCHO was not included when it was being simulated. Using the averaged diurnal profiles reduced modeling uncertainty caused by occasional data flaws and data gaps due to instrument maintenance. The photolysis frequencies were calculated based on TUV v5.2 solar spectra. For each scenario simulation, the box model was
set to spin up for 48 hours.

## 3. Results and Discussion

### 3.1 Ozone pollution spreading into the late-winter haze season

Figure 1 shows the observed exceedance frequency (in days month$^{-1}$) of China's National Ambient Air Quality Standard (NAAQS level-1) for 24-h $PM_{2.5}$ (35 µg m$^{-3}$) and for maximum daily 8-hour (MDA8) ozone (100 µg m$^{-3}$), averaged for 2015-2018 and 2019-2021 in the biggest city (i.e., Xi'an) in the Fenwei Plain in central China (Fig. S1). For the observed $PM_{2.5}$, most exceedances were observed in the heating season from November to March, with over 2/3 of the days in individual months exceeding the 24-h $PM_{2.5}$ standard (Fig. 1a). In contrast, for surface ozone, most exceedances were observed in the warm season from May to August, with approximately 2/3 of the days in individual month exceeding the MDA8 ozone standard (Fig. 1b). Combined, over 2/3 of the days throughout the year (except October) exceeded either the ozone or the $PM_{2.5}$ standard, reflecting the severe air pollution in Fenwei Plain (Fig. 1c), despite the decreasing trend found for $PM_{2.5}$ (Fig. S11). A similar trend was observed for the deweathered ozone and $PM_{2.5}$, suggesting meteorological conditions were not affecting the trend significantly (Fig. S12). Moreover, in terms of exceedance of the NAAQS level-2 standard (i.e., 75 µg m$^{-3}$ for 24-h $PM_{2.5}$ and 160 µg m$^{-3}$ for MDA8 ozone), similar pollution patterns were observed, although the exceedance frequency was less due to the relatively loose standard (Fig. S13).

Compared to 2015-2018, the observed exceedances of $PM_{2.5}$ standard during the warm season in 2019-2021 decreased largely e.g., by over 10 days in August (Fig. 1a), while the differences in the observed exceedances of $PM_{2.5}$ standard were only marginal during the heating season in 2019-2021, i.e., less than 3 days for an individual month in winter. In particular, the exceedance for January in 2019-2021 even increased by approximately 1 day (Fig. 1a). Compared to $PM_{2.5}$, the difference in the observed exceedances of the ozone standard between 2015-2018 and 2019-2021 was small, with values mostly below 5 days month$^{-1}$ (Fig. 1b). In contrast to the slightly decreasing trend of $PM_{2.5}$, the observed exceedances of ozone standard even increased by 1-4 days month$^{-1}$ for January and February in 2019-2021 when compared to 2015-2018 (Fig. 1b and Fig. S14). Therefore, while the winter particulate pollution does not appear abating in recent years in terms of frequency of exceedance of $PM_{2.5}$ standard, the spread of ozone pollution into the late winter (i.e., January and February) may aggravate the air pollution issue in the Fenwei Plain.

Figure 2 shows the observed and deweathered (using a machine learning-based deweathering technique; see Method section) diurnal cycle of $PM_{2.5}$ and $O_3$ in late winter (January-February) averaged over 2015-2018 and 2019-2021. For the observed $PM_{2.5}$, elevated concentrations (>100 µg m$^{-3}$) were observed from the evening to the next noon (20:00 - 12:00; Fig. 2a). The elevated concentration in the evening and morning was due to the primary emission from various sources and secondary particulate formation (discussed in Sect. 3.3), coupled with a shallow boundary layer (Fig. S15) and the topography of the Fenwei Plain (Cao and Cui, 2021), favoring the build-up of air pollutants. However, $PM_{2.5}$ started to decrease at 12:00, reaching the lowest concentrations (<100 µg m$^{-3}$) at 16:00-18: 00. The decrease in the afternoon was likely due to the reduced primary emission and increased boundary layer height (BLH). The increasing BLH in the afternoon diluted the air pollutants, which was confirmed by the decreased CO at the same time (Fig. S16). The $\Delta CO$ corrected $PM_{2.5}$ first showed an increase at noon and decreased afterward (Fig. S16) probably because the $PM_{2.5}$ emissions and/or secondary formation were insufficient to counter the diluting effects in the afternoon. Compared to 2015-2018, the observed $PM_{2.5}$ in 2019-2021 showed a larger decrease (10 µg m$^{-3}$) in the late evening and morning, while the decrease in the afternoon was less significant (~5 µg m$^{-3}$; Fig. 2). A similar trend was also observed for the deweathered $PM_{2.5}$, suggesting the reduction in $PM_{2.5}$ in recent years was mainly caused by emission reduction.

Different from $PM_{2.5}$, both the observed (Fig. 2b) and deweathered $O_3$ (Fig. 2d) showed small differences between 2019-2021 and 2015-2018 during the morning hours (00:00-12:00; sunrise at 8:00) when $PM_{2.5}$ showed the largest reduction during the same period. In contrast, the increase in $O_3$ concentration in 2019-2021 was most prominent in the afternoon when $PM_{2.5}$ reduction was less prominent. Given that the observed $NO_2$ also showed the largest reduction (20 µg m$^{-3}$, Fig. S17) in the

afternoon, the increase in $O_3$ in the afternoon for 2019-2021 was likely caused by the reduced titration effects, coupled with the reduced $PM_{2.5}$ that scavenged less $RO_x$ ($OH+HO_2+RO_2$) radicals. This is consistent with the VOC-limited regime for the $O_3$ formation in the Fenwei Plain for 2019-2021 (discussed in Section 3.4). In particular, for a VOC-limited regime, a moderate decrease in $NO_x$ emission is accompanied by an increase in $O_3$.

**3.2 OVOCs and ozone formation potential in the winter: a case study**

To understand the formation processes of ozone in winter, precursor gases of VOCs and OVOCs were measured using a Vocus-PTR in Xi'an from 24 January 2021 to 6 February 2021 (Sect. 2.1). Table 1 shows the top 10 VOC/OVOC species that were quantified, along with the ozone formation potential (OFP) that was estimated using the maximum incremental reactivity (MIR) (Carter, 2010). Formaldehyde (HCHO) was the most dominant OVOC, with a mean concentration of 4.16 ppb. Its ozone formation potential (OFP) was also the highest (54.4 µg m$^{-3}$), reflecting its high reactivity. The important role of formaldehyde in ozone formation is consistent with the findings in the North China Plain (Li et al., 2021). Note that formaldehyde can be directly emitted from incomplete combustion of e.g., biomass and fossil fuel, while it can also be produced from the oxidation of reactive VOCs (Su et al., 2019).

By constraining ozone and other VOCs/OVOCs (except formaldehyde) in a 0-D box model, the secondary formation of formaldehyde, fueled by photochemical reaction, was simulated (Fig. 3). Compared to the mixing ratio of the measured formaldehyde, the model reproduced the measured formaldehyde at a similar level of around 4.2 ppb. However, compared to the simulated formaldehyde, the elevated concentration of the observed formaldehyde in the evening and morning (22:00 – 8:00; Fig. 3) was likely associated with the primary emission from biomass and fossil fuel combustion, coupled with a shallow boundary layer (Fig. S15). Note that the primary emission of formaldehyde as well as the regional transport was not considered in the box model. The modelled formaldehyde showed an increased mixing ratio at noon (Fig. 3a), corresponding to the increased local formation as indicated by the increased solar radiation and OH radical (Fig. 3b). This is because the box model only considered photochemical formation, not taking into account the scavenging effects of $RO_x$ ($OH+HO_2+RO_2$). High levels of $PM_{2.5}$ (> 80 µg m$^{-3}$) were observed in the morning (Fig. 2), which were likely playing an important role in scavenging $RO_x$ radicals (Shao et al., 2021). Therefore, the modelled spike in formaldehyde at noon was not observed. Similarly, the highest concentration of ozone was coincident with the lowest level of $PM_{2.5}$ in the late afternoon (15:00-17:00) likely due to the reduced scavenging effects which were associated with reduced $PM_{2.5}$ concentration. As a comparison, the highest level of downward solar radiation or the estimated OH radical was in midday (12:00-14:00), corresponding to the highest ozone production rate simulated in the box model (Fig. S18).

Acetone was the second most abundant OVOC, with a mean mixing ratio of 3.3 ppb. However, due to the low reactivity of acetone, its OFP was less significant (3.2 µg m$^{-3}$). Acetone was also found to be one of the most abundant OVOC in a summer campaign conducted in Xi'an, although the acetone mixing ratio in this study was at the lower end of the concentration ranges (2.1-6.6 ppb) sampled in summer 2019 (Song et al., 2021). Nitrogen-containing VOCs of acetonitrile (mean: 3.0 ppb) and formamide (1.1 ppb) were also abundant. However, these nitrogen-containing VOCs were generally considered minor contributors to ozone formation (Carter, 2010), and their OFPs were not evaluated here. Other important ozone precursors of include butanedione (1.6 ppb), acetaldehyde (1.2 ppb), and methyl furans (0.65 ppb).

Aromatics including benzene (1.2 ppb), phenol (0.85 ppb), and toluene (0.76 ppb) were also the top VOCs or OVOCs, contributing to OFP of 3.0 µg m$^{-3}$, 9.9 µg m$^{-3}$, 12.6 µg m$^{-3}$, respectively (Table 1). Toluene (0.76 ppb) was largely reduced when compared to the values (1.9-10.5 ppb) reported previously e.g., in autumn to winter 2017 (Li et al., 2022), although the concentrations were not directly comparable given the different sampling techniques and time. The large reduction was partly due to the reduction in emission intensities. It is important to note that, without the high mass-resolution Vocus-PTR, toluene can be largely overestimated due to the dominant ion ($C_3H_8O_3H^+$) at the same nominal mass to charge ($m/z$) ratio as toluene

i.e., at $m/z$ 93 (Fig. S19). In this study, aromatics were estimated to contribute to 17% of total OFP, slightly higher than its mass fraction (16%) in VOCs.

**3.3 An increased role of secondary particulate formation in winter haze**

To understand the changes in $PM_{2.5}$ and organic aerosol (OA) sources in the winter season, the chemical composition of non-refractory particulate matter (NR-PM) and the corresponding OA factors were compiled (Fig. 4 and Table S3) based on the available studies conducted at the same sampling site in urban Xi'an using an on-line AMS or ACSM (Table S4). Compared to the winter mean NR-PM concentration (261.2 $\mu$g m$^{-3}$) in 2012-2014, a reduction of over 60% in the NR-PM concentration (100.6 $\mu$g m$^{-3}$) in 2018-2021 was observed (Fig. 4). Such a large reduction was mainly due to the reduction in primary particulate emissions including primary OA and chloride, as well as the reduction in the precursor gases of e.g., $SO_2$ that can form secondary species of sulfate, starting from the implementation of the clean air act in 2013 and the "Blue Sky Protection Campaign" in 2018. For example, the city-wide averaged concentration of CO, a surrogate of primary emission due to the incomplete combustion of fossil and biomass burning, showed a similar level of (60%) reduction from 2015 to 2021 (Fig. S20). For the same period, an approximately 60% reduction in city-wide $SO_2$ was also seen where $SO_2$ is mainly associated with fossil fuel combustion (Fig. S20). Despite the large reduction in NR-PM concentrations, exceedances of NAQQS standards were still frequently observed (see Sect. 3.1).

In terms of chemical composition, OA fraction in NR-PM remained similar, accounting for approximately half of the NR-PM for both periods (Fig. 4a), although its absolute concentration decreased by 62% from 117.1 $\mu$g m$^{-3}$ in 2012-2014 to 44.6 $\mu$g m$^{-3}$ in 2018-2021. The reduction in OA concentration was mainly due to the large reduction in primary OA while oxygenated OA remains at a similar level (Fig. 4b). Specifically, the primary fossil fuel OA factor was reduced by 77% (28.9 $\mu$g m$^{-3}$), while biomass burning OA was reduced by 72% (24.2 $\mu$g m$^{-3}$). The reduction in fossil fuel and biomass burning was consistent with the reduction in CO and $SO_2$ as discussed above. Similarly, the primary cooking OA factor was also largely reduced by 84% (16.4 $\mu$g m$^{-3}$). The reduction in cooking emissions could be partly due to upgraded kitchen facilities (Liu et al., 2022), although we note that the sampling time and duration of the compiled studies were not the same (Table S3), causing uncertainties in the evaluations of changes in OA factors years. Future studies with long-term continuous measurements e.g., using ACSM (Chen et al., 2022), will improve the understanding of trends in OA factors in this region. Nevertheless, compared to POA factors, OOA was marginally reduced by 11% (3.0 $\mu$g m$^{-3}$). Due to the larger reduction in primary OA factors, the OOA fraction accounted for over half (52%) of the total OA in 2018-2021, twice higher in terms of fractional contribution compared to the value (22% of OA) in 2012-2014. Biomass burning OA (21% of OA) and fossil fuel OA (20% of OA) had similar contributions to the total OA (Fig. 4b) in 2018-2021. The increased fractional contribution of OOA in recent years was likely due to the emission of its precursor VOCs that were not decreasing in the Fenwei Plain and the increased oxidizing capacity as indicated by the increased ozone concentration. The increase in secondary OA was consistently observed in other areas in China (Li et al., 2022; Nie et al., 2022), which may change of volatility and viscosity of OA (Feng et al., 2023).

Among the inorganic components of NR-PM, chloride showed the largest reduction (84% or 22.9 $\mu$g m$^{-3}$) in the 2018-2021 winter when compared to that in 2012-2014 (Fig. 4a). Chloride was mainly associated with the primary emission from coal and biomass burning. The large reduction in chloride was consistent with the reduction in the primary fossil fuel and biomass burning OA factors (Fig. 4b). Sulfate showed a reduction of 61% (32.3 $\mu$g m$^{-3}$), while nitrate showed a reduction of 50% (17.3 $\mu$g m$^{-3}$). Ammonium showed a reduction of 51% (10.8 $\mu$g m$^{-3}$). The reduction in sulfate was consistent with the reduction in its precursor $SO_2$ (Fig. S20). Because nitrate reduction was less significant, nitrate increased its fractional contribution (17% of NR-$PM_{2.5}$) in 2018-2021 from 13% of NR-$PM_{2.5}$ in 2012-2014. Compared to $SO_2$, the reduction in $NO_2$ was only observed in recent years from 2018 to 2021, while $NO_2$ showed a slight increase from 2015 to 2018 (Fig. S20). Therefore, nitrate is likely to increase its fractional contribution to the total $PM_{2.5}$ mass because $NO_2$ was not decreasing as significantly as $SO_2$. Also, the increased oxidizing capacity leads to enhanced nitrogen oxidation ratio and thus enhanced nitrate yield.

**3.4 Impact of control policies on the ozone and secondary aerosol formation: observation-based modelling**

As a response to the coronavirus pandemic (i.e., COVID-19), the lockdown measures, which were implemented in the Fenwei Plain from 24 January to 28 February 2020, provided a unique opportunity to investigate the impact of strict emission control on the ozone and SOA formation potential in comparison to that before (2019) and after the pandemic (2021). Figure 5 shows the vertical column density of $NO_2$, a surrogate of anthropogenic emission, and formaldehyde, a surrogate of reactive OVOCs, over the same period (24 January – 28 February) for three consecutive years from 2019 to 2021. Compared to $NO_2$ (18-25 $\times10^{15}$ molecules cm$^{-2}$) before the pandemic in 2019, the mean vertical column density of $NO_2$ during the lockdown (6-10 $\times10^{15}$ molecules cm$^{-2}$) in Xi'an 2020 reduced by up to over 70%. However, without the lockdown measures in 2021, the $NO_2$ column density bounced back quickly, but with a slightly lower density in the city center of Xi'an (16-20 $\times10^{15}$ molecules cm$^{-2}$). In particular, the $NO_2$ in other cities in the Fenwei Plain was only 70-80% (or 14-20 $\times10^{15}$ molecules cm$^{-2}$) of that in 2019 (Fig. 5).

The large reduction in $NO_2$ during the lockdown period in 2020 implies a sharp reduction of anthropogenic emissions. However, different from $NO_2$, the vertical column concentration for formaldehyde remained at the same level for the lockdown year in 2020 when compared to before the lockdown in 2019 with densities in the range of 8-10 $\times10^{15}$ molecules cm$^{-2}$ in urban Xi'an (Fig. 5), suggesting a weaker reduction of the reactive OVOC despite the significant reduction in anthropogenic activities including traffic and some industries that were not operative during the pandemic in 2020. In 2021, the formaldehyde densities even increased a little bit when compared to 2019 (Fig. 5). Due to the large ozone formation potential from formaldehyde (as discussed in Sect. 3.3) and the reduced $NO_x$ emissions (titration effects), surface ozone concentrations increased its NAAQS exceedance frequency in recent winters as demonstrated in Fig. 1. This is also consistent with the fact that ozone formation pathways in the city centers of the Fenwei Plain are in the VOC-limited regime as corroborated by the low TROPOMI HCHO/$NO_2$ ratios with values below 1 (Fig. S21). The low HCHO/$NO_2$ values in the Fenwei Plain were consistent with that observed in most Chinese cities where $O_3$ was in the VOC-limited region (Li et al., 2022; Tan and Wang, 2022).

To examine the response of in situ photochemistry to up to 70% reduction in $NO_2$ as the scenario of the strict emission control during the lockdown, simulations of the changes in the mean rate of self-reaction between peroxy radicals were performed using a box model. Figure 6 shows that the mean self-reaction rate between peroxy radicals increase by 12-75% in response to the reduction of 20-70% of $NO_2$ while the VOCs were constant. Self-reaction between peroxy radicals includes the autooxidations of $RO_2$ ($RO_2$ + $RO_2$) and $HO_2$ ($HO_2$ + $HO_2$) and cross reactions ($RO_2$ + $HO_2$), some of which lead to SOA formation (Zhao et al., 2018). Given that self-reaction between peroxy radicals is an important pathway for SOA formation potential, the increase in self-reaction rate can partly explain the increase in SOA formation potential in recent winters (Fig. 3). In addition to the self-reaction, some of the reaction between $NO_x$/$NO_3$ and peroxy radicals can also produce organic nitrate that could partition to the particle phase (Lin et al., 2021). However, such formation pathways were likely playing a less important role when NOx were largely reduced. In a scenario of a further 20% reduction in VOCs, self-reaction rate was reduced as compared to the reduction of $NO_2$ alone. This is because reducing VOCs reduces the availability of peroxy radicals, while the competition between peroxy radicals and NOx was also reduced at the same time. However, the rate is still 14% higher than the base case. A large reduction (50-80%) in the self-reaction rate can be only achieved in response to a further 50-70% reduction of VOCs as compared to the base case. The modeling results imply the co-benefits of controlling VOCs in reducing ozone and SOA pollution.

**3.5 Atmospheric implication**

The increase of ozone pollution in winter, concurrent with haze pollution in the Fenwei Plain is concerning since ozone pollution is generally considered a summertime problem (Li et al., 2019; Li et al., 2022). To make things worse, the increase in ozone pollution would stimulate the formation of secondary $PM_{2.5}$, including SOA, nitrate and sulfate, and, therefore, partly negating the efforts to reduce haze pollution in winter. This is demonstrated in this study as the secondary pollutants, and in

particular, SOA and nitrate, are becoming increasingly important in recent winter haze. Due to the unique topology of the Fenwei Plain (Cao and Cui, 2021), the air pollutants, both locally emitted/produced and regionally transported, are favorably trapped inside the basin on winter night when the boundary layer height is less than 100 m, accumulating air pollutants to health-threatening level. As the development of boundary layer after sunrise, ozone formation is increased in the late afternoon. As a result of the increased $O_3$, secondary particle pollutants are further increased and subsequently accumulated at night. Such a feedback loop makes it more challenging to reduce ozone and $PM_{2.5}$ in the Fenwei Plain than in other regions e.g., North China Plain. If inappropriate emission control measures are implemented, one may expect more frequent ozone-$PM_{2.5}$ pollution in winter.

The increase of ozone in winter haze is explained by the in-situ photochemical reaction caused by the reduction of $NO_x$, while reactive OVOCs remain constantly elevated. The abrupt decrease of $NO_x$, as observed during the lockdown in 2020, provides real-world evidence of the side effects of implementing inappropriate emission-control measures, targeting $NO_x$ but letting loose VOCs. Here, we also show that $NO_2$ levels without the lockdown in 2021, were still 10-20% reduced when compared to before the pandemic in 2019, while the reactive HCHO was even higher. Such a trend needs to be reversed in order to achieve the synergic reduction of $O_3$ and $PM_{2.5}$.

The complexity of the $O_3$-$PM_{2.5}$ interaction also needs to be considered because $PM_{2.5}$ scavenges $HO_2$ radicals that would otherwise produce $O_3$. Such an interaction is more important when the starting $PM_{2.5}$ concentration is high (>40 μg m$^{-3}$) (Shao et al., 2021). Here, we show that, although the city-wide mean $PM_{2.5}$ concentration decreased in recent years, the $PM_{2.5}$ concentration is still above 80 μg m$^{-3}$ for both observed and de-weathered $PM_{2.5}$, i.e., still in the range of $PM_{2.5}$ concentration where the reduction of $PM_{2.5}$ can increase $O_3$ due to the reduced scavenging effect. Since the reduction in $PM_{2.5}$ in Xi'an is slower (-3.3% yr$^{-1}$) (Wang et al., 2022) than in other areas, e.g., Beijing (-7.8% yr$^{-1}$) (Vu et al., 2019) and Shanghai (-6.4% yr$^{-1}$) (Wang et al., 2022), a large increase in $O_3$ in winter is expected as $PM_{2.5}$ is only reduced at a similarly slow rate in the near future. Only with a more aggressive reduction of the precursor gases can we achieve the goal of synergic reduction $O_3$ and $PM_{2.5}$ pollution in Fenwei Plain.

## 4. Conclusion

In this study, an increase in $O_3$ pollution in the winter haze was revealed through the analysis of city-wide mean concentrations of air pollutants for 7 years (2015-2021). The increased $O_3$ was not due to the changes in meteorological variations, as confirmed by the trend of deweathered $O_3$, but due to the constantly elevated reactive OVOCs, and in particular formaldehyde in combination with the reduced $NO_2$. An increase in $O_3$ stimulated the formation of secondary aerosols, including SOA and nitrate, which were shown to increase their fractional contributions to the total $PM_{2.5}$. A further extension of ozone pollution in winter haze is expected if $NO_x$ and $PM_{2.5}$ were reducing at the current rate, while reactive OVOC were not reducing. Through scenario analysis in a chemical box model, we show the co-benefits of reducing $NO_x$ and VOCs simultaneously in reducing ozone and SOA. The complex interaction between $O_3$ and $PM_{2.5}$, coupled with the topology of the Fenwei Plain and the evolution of the boundary layer height, highlight the challenges in further reducing particulate pollution in winter despite years of efforts to reduce emissions. A synergic reduction in $PM_{2.5}$ and $O_3$ pollution can only be achieved through a more aggressive reduction of their precursor gases.

## ASSOCIATE CONTENT

**Data availability.** All data needed to evaluate the conclusions in the paper are present in the paper and/or the Supplement. Also, all data used in the study are available from the corresponding author upon request.

**Supporting Information**

Supplementary Figures (Fig. S1-S21) and Table (Table S1-S4)

AUTHOR INFORMATION

**Corresponding author**

Ru-Jin Huang (rujin.huang@ieecas.cn)

**Author Contributions**

CL and RJH designed the study. CL, JD, and HBZ conducted measurements, data analysis, and source apportionment. CL prepared the manuscript with contributions from all co-authors.

**Competing interests.** The authors declare that they have no conflicting interests.

**ACKNOWLEDGMENT**

This work was supported by National Natural Science Foundation of China (NSFC) under Grant No. 42277092, 42107126, 42207137, and 41925015, the Strategic Priority Research Program of Chinese Academy of Sciences (No. XDB40000000), the Chinese Academy of Sciences (no. ZDBS-LY-DQC001), Institute of Earth Environment (E051QB2837), and the Cross

Innovative Team fund from the State Key Laboratory of Loess and Quaternary Geology (No. SKLLQGTD1801).

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

**Table 1. Top 10 volatile organic compounds (VOCs)/oxygenated VOCs (OVOCs) and the ozone formation potential (OFP).**

| | Formula | Exact Mass | Assignment | Mean (ppb) | OFP ($\mu g\ m^{-3}$) |
|---|---|---|---|---|---|
| 1 | HCHOH | 31.017841 | Formaldehyde[a] | 4.16 | 54.4 |
| 2 | C3H6OH | 59.049141 | acetone | 3.39 | 3.2 |
| 3 | C2H3NH | 42.033825 | acetonitrile | 3.03 | - |
| 4 | C4H6O2H | 87.044055 | butanedione | 1.63 | 27.7 |
| 5 | C2H4OH | 45.03349 | acetaldehyde | 1.20 | 15.8 |
| 6 | C6H6H | 79.054226 | benzene | 1.19 | 3.0 |
| 7 | CH3NOH | 46.028740 | formamide | 1.10 | - |
| 8 | C6H6OH | 95.049141 | phenol | 0.85 | 9.9 |
| 9 | C7H8H | 93.069876 | toluene | 0.76 | 12.6 |
| 10 | C5H6OH | 83.04914127 | methylfurans | 0.65 | 20.0 |
| Total | | | TVOCs | 18.0 | 146.8 |

[a]Formaldehyde concentration was scaled using the vertical column concentration.

**a** Observed exceedances of PM$_{2.5}$ standard (days month$^{-1}$)

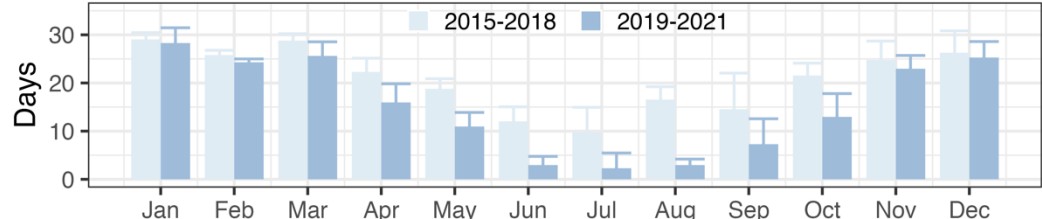

**b** Observed exceedances of Ozone standard (days month$^{-1}$)

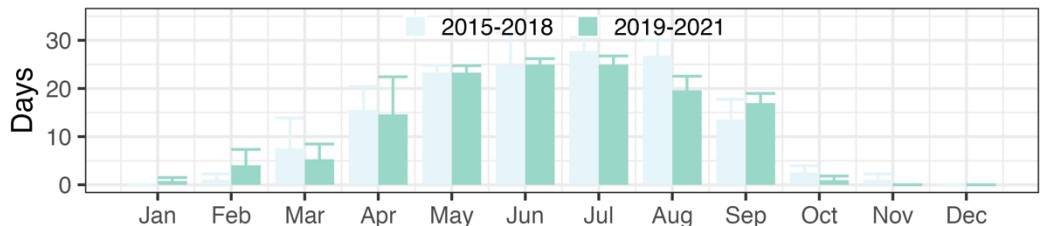

**c** Observed exceedances of Ozone or PM$_{2.5}$ standard (days month$^{-1}$)

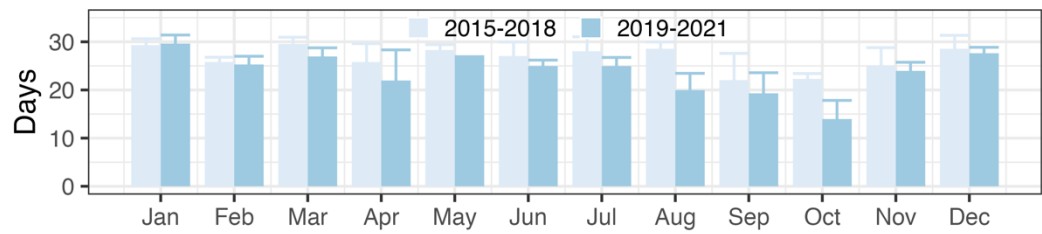

**Figure 1. Observed exceedance frequency (in days month$^{-1}$) of PM$_{2.5}$ (a); Ozone (b);** Ozone or PM$_{2.5}$ **(c) standard (NAAQS level-1) in the biggest city (i.e., Xi'an) in Fenwei Plain, averaged over 2015-2018 and 2019-2021. Ozone and PM$_{2.5}$ were averaged from 13 monitoring sites in Xi'an. Error bar represents one standard deviation.**

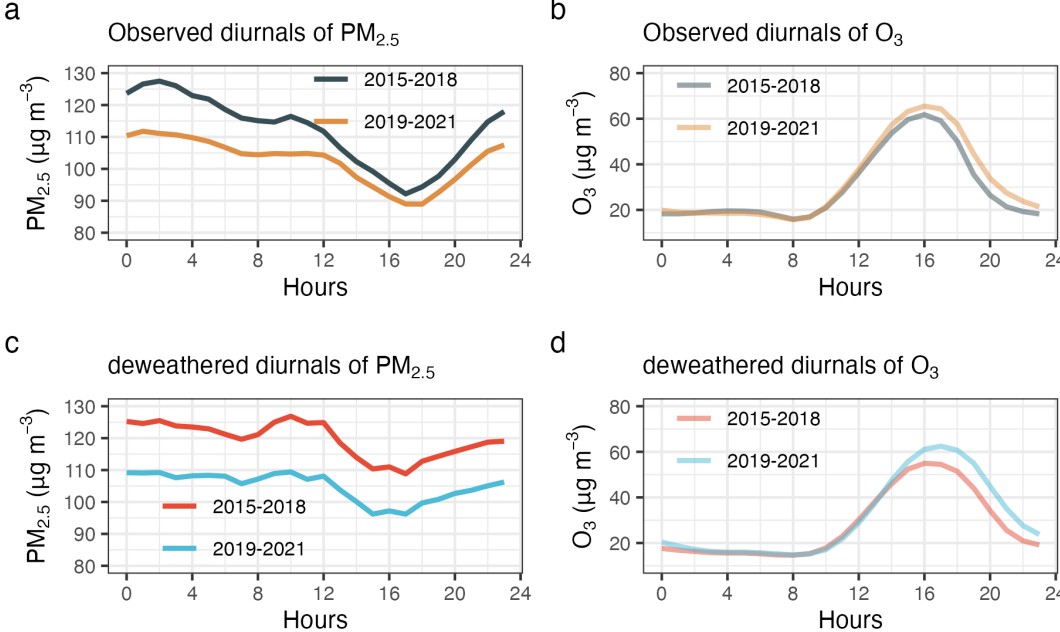

**Figure 2. Observed and deweathered diurnal patterns for PM$_{2.5}$ (a,c) and O$_3$ (b, d) for January-February in 2015-2018 and 2019-2021**

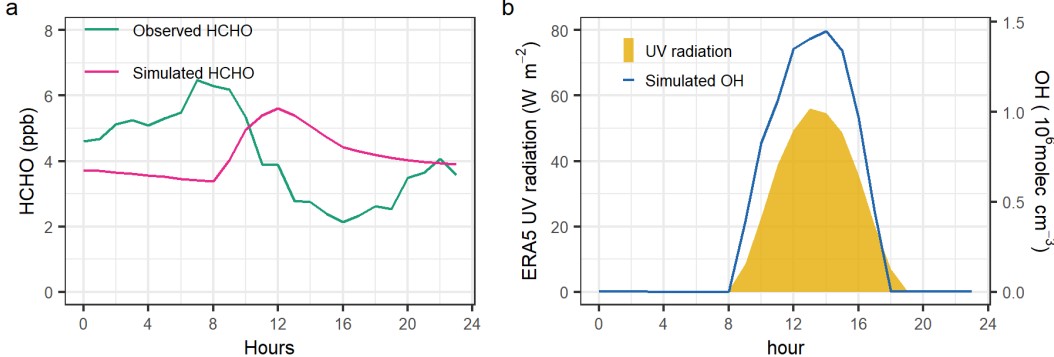

**Figure 3. (a) Observed and simulated formaldehyde (HCHO) and (b) the EAR5 UV radiation flux and the simulated OH radical in the box model.**

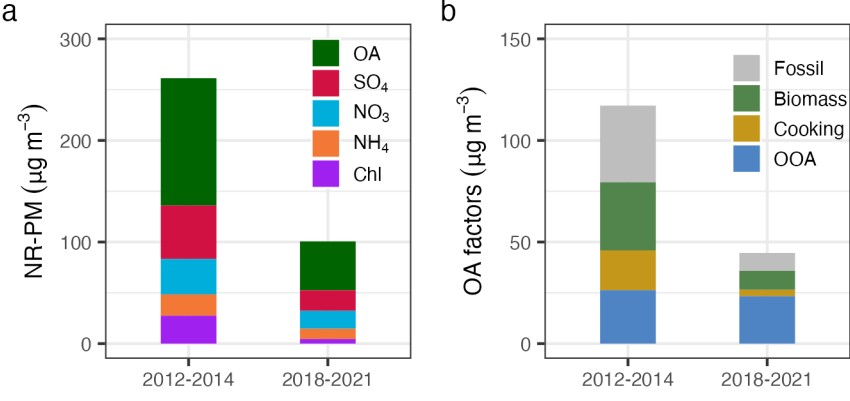

Figure 4. (a) Mean chemical composition of non-refractory particulate matter (NR-PM$_{2.5}$) and the OA factor for the winter in 2012-2014 and 2018-2021.

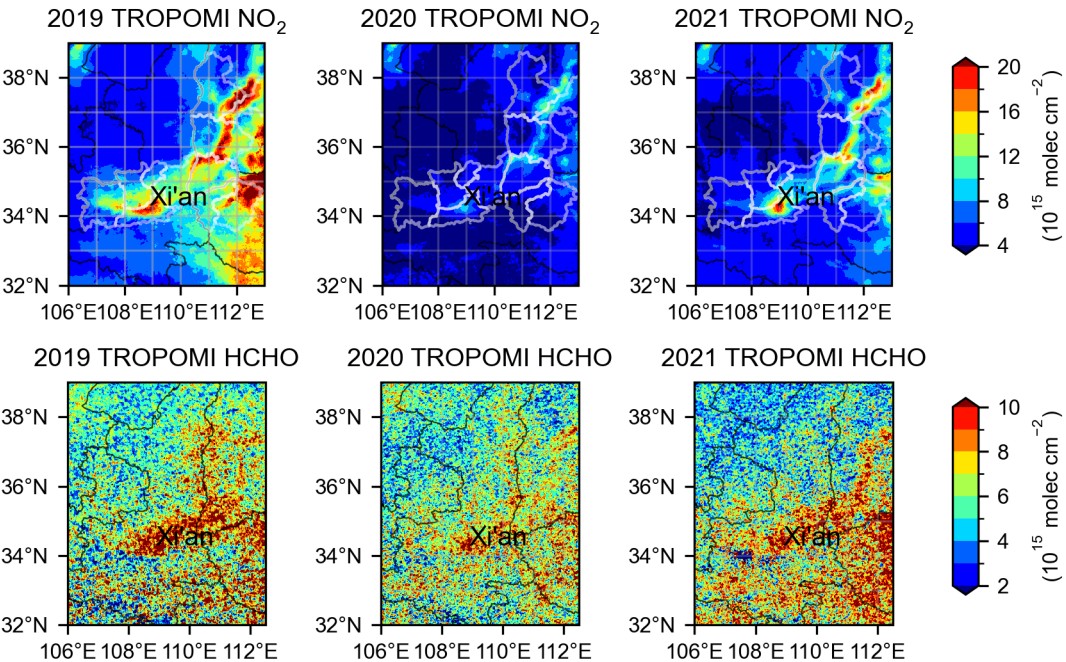

Figure 5. Tropospheric NO$_2$ column densities, a surrogate of anthropogenic emissions, and tropospheric HCHO column densities, a surrogate of OVOCs, measured by the TROPOMI satellite instrument in January-February in 2019, 2020, and 2021.

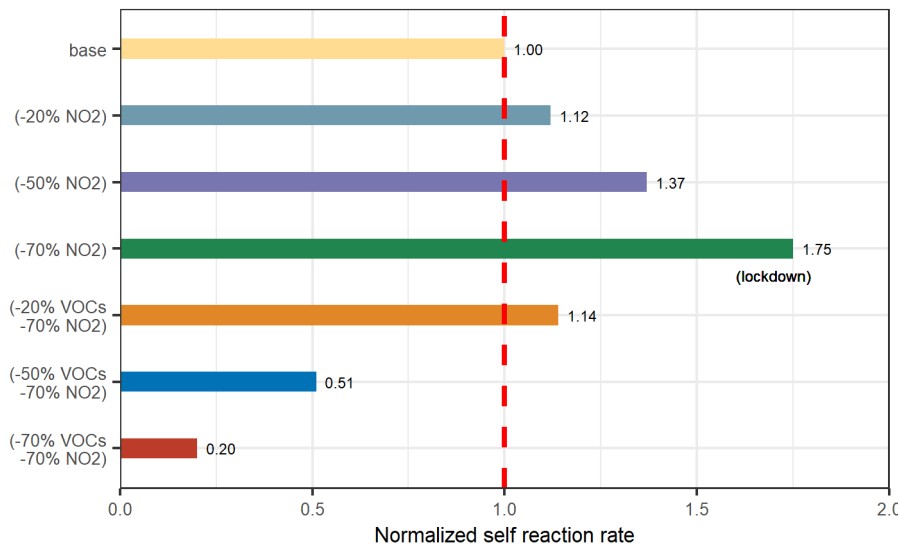

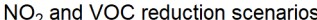

**Figure 6. Changes in self-reaction rate between peroxy radicals with 20-70% reduction in NO₂ and a further 20-70% in VOC reduction. Covid-19 lockdown in 2020 correponded to a reduction of NO₂ of 70%. The reaction rate in different reduction scenarios was normalized to the value in the base run (guided by the red dash line).**

565