# Peer review of "Elucidating ozone and PM2.5 pollution in Fenwei Plain reveals the cobenefits of controlling precursor gas emissions in winter haze"

_EGUsphere, 2022_

## Author Comment (AC1)

We thank the reviewers for the careful comments that helped improve the manuscript. We have revised the manuscript accordingly. Below is the point-to-point response to each comment, where our responses are in blue.

Reviewer#1

General comments:

The paper titled "Elucidating ozone and PM2.5 pollution in Fenwei Plain reveals the co-benefits of controlling precursor gas emissions in winter haze" by Lin et al. evaluates the status of ozone and PM2.5 pollution in a typical megacity of the Fenwei Plain, one of the most polluted regions in China, which reported a general trend of increasing secondary pollution (ozone and SOA) in winter haze, and the causes of this trend and the possible measures in controlling the complex pollution by O3 and PM2.5 were further studies and discussed. With this, the authors claimed that the co-benefits of reducing NOx and VOCs simultaneously in reducing ozone and SOA, that would be also suitable for other polluted regions of China suffering ozone and PM2.5 currently. The manuscript was well written and presented clearly. Therefore I recommend the publication of Lin et al. work after some issues were properly revised and improved.

Response: We thank the reviewer for the positive comments. We provide a point-to-point response to each comment below.

Specific and technical comments:

Method, more details in the calibration of PTR-MS should be provided. In addition, What kinds of VOCs species were used in the standard mixture? Please list the VOCs species that calculated from the kinetic rate constant, and the uncertainty on the calculated VOCs should be discussed.

Response: We have now provided more details regarding the Vocus-PTR-MS calibration. We have listed all the VOC species that were used for calibration in the revised Table S1. VOC species with the kinetic rate constant was listed and the uncertainties on the calculated VOCs were discussed.

In the revised Section 2.1, it now reads, "…Vocus-PTR was calibrated with VOC/OVOC standard mixture (Table S1). The sensitivity of Vocus-PTR towards uncalibrated VOC compounds was calculated from the kinetic rate constant following Krechmer et al. (2018). Uncertainties for the calibrated VOC/OVOC species were 15%, while the uncertainty was 30% for the uncalibrated VOC compounds (Table S2) with known reaction rate constant ($k_{PTR}$) (https://kb.tofwerk.com/tofware/; last access: 1 March 2023). Due to the low transmission efficiency for low-mass molecules that caused high uncertainties in quantification, formaldehyde (HCHO) was scaled to the mean concentration of the surface HCHO based on satellite observation using the empirical relationship (Zhang et al., 2012)…"

Table S1. A list of the VOC species, the proton transfer reaction rate coefficients ($k_{PTR}$) between the hydronium ion ($H_3O^+$) and selected VOCs, and the obtained sensitivities during calibration.

| VOC species | $k_{PTR}$ ($10^{-9}$ molec cm$^{-3}$ s$^{-1}$) | Sensitivity (cps ppbv$^{-1}$) |
|---|---|---|
| Benzene | 1.93 | 2,351 |
| Toluene | 2.08 | 2,446 |
| m-Xylene | 2.27 | 3,066 |
| 1,2,4-Trimethylbenzene (TMB) | 2.4 | 2,835 |
| Acetone | 3.44 | 6,773 |
| Methyl ethyl ketone (MEK) | 3.39 | 5,191 |
| Acetonitrile | 4.2 | 1,481 |
| Acetaldehyde | 3.24 | 1697 |

Line 109-112, it is better to provide more details for NR-PM2.5 monitored by an AMS which usually measured NR-PM1. I note that a novel PM2.5 was firstly equipped with AMS for the winter campaign in 2014 (Elser et al., 2014), It is unclear for the other winter campaigns.

Response: We agree and have provided more details for NR-PM$_{2.5}$ monitored by the deployed AMS with a PM$_{2.5}$ inlet in the revised manuscript.

Elser et al., (2016) first applied a PM$_{2.5}$ ToF-AMS in Xi'an in 2014 and evaluated the performance of PM$_{2.5}$ inlet comprehensively. Later at the same sampling site in 2019 and 2022, an LToF SP-AMS with a PM$_{2.5}$ inlet was deployed with satisfactory performance as detailed in Duan et al. (2021). The deployed PM$_{2.5}$ inlet shared the same design as that in Elser et al. (2016). The time series of the measured NR-PM$_{2.5}$ concentrations were in good agreement with the PM$_{2.5}$ monitored at a nearby environmental monitoring site (see Fig. S1 in Duan et al. (2021) and Fig. S3 in Lin et al. (2021))

In the revised Section 2.1, it now reads, "For AMS measurement, Elser et al. (2016) first deployed an AMS with a PM$_{2.5}$ inlet and comprehensively evaluated its performance during the winter campaign in Xi'an. Based on the same design, a soot particle AMS (SP-AMS) equipped with a PM$_{2.5}$ inlet was recently deployed at the same sampling site, quantifying PM$_{2.5}$ chemical composition with high mass and temporal resolution (Lin et al., 2021; Duan et al., 2022)."

Line 121-122, why the reduction in NO2 for the observation sites was not used? Which would be more precisely than the satellite image.

Response: We agree that site observations of NO$_2$ were more precise than satellite images. In this study, we have simulated a range of NO$_2$ reduction scenarios with a reduction of 20-70%. This range of reduction scenarios covered the lower and upper limits of NO$_2$ reduction from observation sites and satellite images. To avoid confusion, this sentence has been removed.

Line 144-146, please list the VOC/VOCs information that used as input data for box model. I note that HCHO was not used to constrain the model, how about the other OVOCs? Considering the OVOCs was also from secondary formation. In addition, I am concerns on the model performance in the ozone simulations, as the majority of alkanes was unavailable in the model if only the VOC/VOCs measured by the PTR-MS. As least, the authors should provided more details in the performance of the box model and the analysis in the uncertainty.

Response: We have now added Table S2 to list all the VOC/OVOCs information that was used as input in the box model. Details of OVOCs other than HCHO were also shown in Table S2.

The performance and uncertainties of the box model are now discussed in the revised Sect 2.4. It now reads, "…Surface observational data were averaged for the overlapping period with OVOC measurements to get the diurnal profiles of each pollutant. The averaged air pollutants, including $NO_2$, $O_3$ (not included when it was simulated, Fig. S10), CO, and the measured VOC/OVOCs (Table S2) were used to constrain the model at hourly resolution (base run). The box model reproduced the measured ozone reasonably well (Fig. S10) after optimizing the correction of the simulated photolysis rate. However, due to the uncertainties of the box model which did not consider factors like regional transport and heterogenous chemistry, as well as the parameterized ozone deposition, the modeling results were only used to understand the atmospheric chemistry involving OVOCs by comparing different scenarios. Note that HCHO was not included when it was being simulated…"

Table S2. A list of VOC/OVOC species that were included in the box model.

| Formula | Assignment | Mean (ppb) |
| --- | --- | --- |
| HCHOH | Formaldehyde[a] | 4.16 |
| C3H6OH | acetone | 3.39 |
| C4H6O2H | butanedione | 1.63 |
| C2H4OH | acetaldehyde | 1.20 |
| C6H6H | benzene | 1.19 |
| C6H6OH | phenol | 0.85 |
| C7H8H | toluene | 0.76 |
| C3H6O2H | Methyl acetate | 0.35 |
| C4H8OH | Butyraldehyde | 0.29 |
| C4H6OH | MEK | 0.29 |
| C5H9 | Isoprene | 0.23 |
| C3H4O2H | Propanal | 0.13 |
| C4H6OH | MVK | 0.10 |
| C3H4OH | Acrolein | 0.10 |
| C9H13 | TMB | 0.03 |

[Figure]

Figure S10. Diurnal of the simulated (Sim) and observed $O_3$ during the VOC/OVOC sampling period. The shaded area represents one standard deviation.

Line 200-202, I do not agree that the secondary formation could be the major source of formaldehyde, as the measured and modelled formaldehyde showed different diurnal pattern. The similar level may suggest large uncertainty in the modelled formaldehyde.
Response: We agree. In Sect. 3.2, it now reads, "…Compared to the mixing ratio of the measured formaldehyde, the model reproduced the measured formaldehyde at a similar level of around 4.2 ppb. However, compared to the simulated formaldehyde, the elevated concentration of the observed formaldehyde in the evening and morning (22:00 – 8:00) was likely associated with the primary emission from biomass and fossil fuel combustion, coupled with a shallow boundary layer (Fig. S15)…".

Line 245-247, the significant reduction in primary fossil fuel OA (77%) from 2012-2014 to 2019-2021 could be expected, due to the implementation of the clean air act in 2013. The more magnitude of reduction in cooking OA (84%) is interesting, more evidence should provided and discussed here.
Response: We agree that the reduction in fossil fuel OA was expected. Regarding the reduction in Cooking OA, it could be related to upgraded kitchen facilities, as well as the uncertainties in comparing the changes in COA. We have now discussed the uncertainties in the comparison. In the revised manuscript it now reads, "…The 5 datasets obtained at the same sampling site were averaged to gain insights into the changes in chemical composition and OA factors over these years, although it is noted that measurements were not conducted at the same period in each year with the same duration. The one standard deviation (sd) (Table S4) ranged from 1.8 to 78.5 µg m$^{-3}$, or 13-110% for the NR-PM species and from 0.7 to 29.4 µg m$^{-3}$, or 14-88% for the OA factors. …" and "…The reduction in fossil fuel and biomass burning was consistent with the reduction in CO and SO$_2$ as discussed above. Similarly, the primary cooking OA factor was also largely reduced by 84% (16.4 µg m$^{-3}$). The reduction in cooking emissions could be partly due to upgraded kitchen facilities (Liu et al., 2022), although we note that the sampling time and duration of the compiled studies were not the same (Table S3), causing uncertainties in the evaluations of changes in OA factors years. Future studies with long-term continuous measurements e.g., using ACSM (Chen et

al., 2022), will improve the understanding of trends in OA factors in this region…"

Rewiewer#2
Review on "Elucidating ozone and PM2.5 pollution in Fenwei Plain reveals the co-benefits of controlling precursor gas emissions in winter haze" by Lin et al.
The study looks into the pollution patterns, sources, and formation mechanism of PM2.5 and ozone. The analysis of 7 years data reveals the severity of air pollution. The author found that increased ozone was due to the constantly elevated reactive OVOCs and the reduced NO2, and then stimulated the increase of particle pollution. A 0-box model was applied to investigated the co-benefits of reducing NOx and VOCs simultaneously in reducing ozone and SOA. Finally, the atmospheric implication helps for developing cost-effective mitigation policies in the future. The results are important for the scientific community to increase their understanding of the O3-PM2.5 interaction. The paper is well written, and the literature is broadly cited. I recommended a minor revision before publication.
Response: We thank this reviewer for the positive comments.

Comments:
Line 40: the font size needs to be constant
Response: Corrected.

Line 115: More detail of the 5 datasets should be provided in this paper, including the sampling time in a year. Since the PM2.5 and OA obviously vary in different seasons, the correction or uncertainty analysis should be presented in the paper.
Response: We have now provided more details (Table S3) about the sampling time and have discussed the uncertainties in the revised manuscript (also see the response to reviewer #1).
In the revised text, it now reads, "…The 5 datasets obtained at the same sampling site were averaged to gain insights into the changes in chemical composition and OA factors over these years, although it is noted that measurements were not conducted at the same period in each year with the same duration. The one standard deviation (sd) (Table S4) ranged from 1.8 to 78.5 µg m$^{-3}$, or 13-110% for the NR-PM species and from 0.7 to 29.4 µg m$^{-3}$, or 14-88% for the OA factors. …" and "…The reduction in fossil fuel and biomass burning was consistent with the reduction in CO and SO$_2$ as discussed above. Similarly, the primary cooking OA factor was also largely reduced by 84% (16.4 µg m$^{-3}$). The reduction in cooking emissions could be partly due to upgraded kitchen facilities (Liu et al., 2022), although we note that the sampling time and duration of the compiled studies were not the same (Table S3), causing uncertainties in the evaluations of changes in OA factors years. Future studies with long-term continuous measurements e.g., using ACSM (Chen et al., 2022), will improve the understanding of trends in OA factors in this region…"

**Table S3.** Sampling time and instruments at the same urban sampling site in Xi'an.

| Sampling time | Season | Instrument | References |
|---|---|---|---|
| 2012.11.15 – 2013.2.21 | 2012-2013 winter | $PM_1$-ACSM | Zhong et al. (2020) |
| 2013.12.13 – 2014.1.6 | 2013-2014 winter | $PM_{2.5}$-AMS | Elser et al. (2016) |
| 2018.12.4 – 2019.3.15 | 2018-2019 winter | $PM_{2.5}$-SP-AMS | Duan et al. (2022) |
| 2020.1.18 – 2020.1.31 | 2019-2020 winter | $PM_{2.5}$-SP-AMS | Duan et al. (2021) |
| 2021.1.14 – 2021.2.6 | 2020-2021 winter | $PM_{2.5}$-SP-AMS | This study |

Line 134-136: What is the advantage of developed random forest model compared to previous de-weathered RF model? mean meteorological variables ? To average the meteorological data at a specific time point during each year?

Response: We have now discussed the advantages of our de-weathered RF model. In the revised Sect 2.3. It now reads, "Random Forest (RF) - based meteorological normalization was first introduced by Grange and Carslaw (2019). However, it is not straightforward to investigate the seasonal variation in the trends of de-weathered air pollutants using the proposed normalization method (Grange and Carslaw, 2019). Vu et al. (2019) enhanced the meteorological normalization procedure by repeatedly resampling the meteorological variables for a particular time point within a four-week period and the resampled variables were fed to the developed RF model. The meteorological normalization method proposed by Vu et al. (2019), is widely applied in different sampling sites (Dai et al., 2021; Shi et al., 2021). In this study, we resampled the meteorological variables by averaging the meteorological variables for the same time point across 7 years. Our proposed method was more straightforward and subject to less variation given that meteorological variables varied a lot over a four-week period (e.g., see the comparison between the resampled and observed temperatures in Fig. S9). Also, our method is less time-consuming. Therefore, the meteorological normalization technology assumed the meteorological variables were invariant across the years. In this scenario, the predicted values represent the changes in air pollutant concentrations that were not affected by meteorological variables (i.e., de-weathered pollutants)."

[Figure]

Figure S9. Monthly variation of observed (a) and resampled (b) air temperatures (air_temp in °C) for 2015-2021. The resampled air temperatures were invariant across the years, in contrast to that for the observed temperatures.

Line 142-148: The detailed species of VOC/OVOCs, instead of only top 10 species, should be provided, since these precursors influence the results of the 0-box model. The input data are averaged diurnal profiles or the total time series? If the averaged diurnal data were applied, how the uncertainties of the model change? More importantly, the verification of the 0-box model is missing. For example, the comparison between measured and predicted concentration of ozone, which was the main object in this study.

Response: We have now listed all the VOC/OVOCs that were used as box model input in Table S2 (also see the reply to Reviewer #1). We have compared the measured and predicted Ozone using averaged diurnal profiles as model input. Using the averaged diurnal profiles is expected to reduce the uncertainty caused by occasional data flaws and data gaps due to instrument maintenance.

In the revised Sect 2.4, it now reads, "…Surface observational data were averaged for the overlapping period with OVOC measurements to get the diurnal profiles of each pollutant. The averaged air pollutants, including $NO_2$, $O_3$ (not included when it was simulated, Fig. S10), CO, and the measured VOC/OVOCs (Table S2) were used to constrain the model at hourly resolution (base run). The box model reproduced the measured ozone reasonably well (Fig. S10) after optimizing the correction of the simulated photolysis rate. However, due to the uncertainties of the box model which did not consider factors like regional transport and heterogenous chemistry, as well as the parameterized ozone deposition, the modelling results were only used to understand the atmospheric chemistry involving OVOCs by comparing different scenarios. Note that HCHO was not included when it was being simulated…"

Table S2. A list of VOC/OVOC species that were included in the box model.

| Formula | Assignment | Mean (ppb) |
|---------|------------|------------|
| HCHOH | Formaldehyde[a] | 4.16 |
| C3H6OH | acetone | 3.39 |
| C4H6O2H | butanedione | 1.63 |
| C2H4OH | acetaldehyde | 1.20 |
| C6H6H | benzene | 1.19 |
| C6H6OH | phenol | 0.85 |
| C7H8H | toluene | 0.76 |
| C3H6O2H | Methyl acetate | 0.35 |
| C4H8OH | Butyraldehyde | 0.29 |
| C4H6OH | MEK | 0.29 |
| C5H9 | Isoprene | 0.23 |
| C3H4O2H | Propanal | 0.13 |
| C4H6OH | MVK | 0.10 |
| C3H4OH | Acrolein | 0.10 |
| C9H13 | TMB | 0.03 |

[Figure]

Figure S10. Diurnal of the simulated (Sim) and observed $O_3$ during the VOC/OVOC sampling period. The shade area represents one standard deviation.

Line 167-170: It seen that only 1-2 days difference found between the two period may

not always support the authors' conclusion. What about the ozone exceedance in each year during 2015-2021? Or what about the uncertainty/standard deviation for the averaged data?

Response: Actually, the difference between the two periods was 1-4 days in January and February. We have updated Figure 1 because our previous calculation treated months with no exceedance as "not a number" (nan) that were not included in the calculation.

Ozone exceedance in each year during 2015-2021 is now added as Figure S14 and discussed in the revised text. The one standard deviation (sd) is now included in the updated Figure 1.

It now reads, "…In contrast to the slightly decreasing trend of $PM_{2.5}$, the observed exceedances of ozone standard even increased by 1-4 days month$^{-1}$ for January and February in 2019-2021 when compared to 2015-2018 (Fig. 1b and Fig. S14)…"

**a**   Observed exceedances of $PM_{2.5}$ standard (days month$^{-1}$)

[Figure]

**b**   Observed exceedances of Ozone standard (days month$^{-1}$)

[Figure]

**c**   Observed exceedances of Ozone or $PM_{2.5}$ standard (days month$^{-1}$)

[Figure]

Figure 1. Observed exceedance frequency (in days month$^{-1}$) of $PM_{2.5}$ (a); Ozone (b); Ozone or $PM_{2.5}$ (c) standard (NAAQS level-1) in the biggest city (i.e., Xi'an) in Fenwei Plain, averaged over 2015-2018 and 2019-2021. Ozone and $PM_{2.5}$ were averaged from 13 monitoring sites in Xi'an. Error bar represents one standard deviation.

[Figure]

**Figure S14.** Observed exceedance frequency (in days month$^{-1}$) of Ozone or PM$_{2.5}$ standard (NAAQS level-1) in the biggest city (i.e., Xi'an) in Fenwei Plain from 2015 to 2021. Ozone and PM$_{2.5}$ were averaged from 13 monitoring sites in Xi'an.

Line 175: The reference or topographic map for the topography favoring the build-up of air pollutants should be provided.

Response: We have now added a reference to support our claim. It now reads, "…the topography of the Fenwei Plain (Cao and Cui, 2021), favoring the build-up of air pollutants…."

Line 177-178: The BLH highlights very obvious difference between afternoon and other time period. The author could correct the PM2.5 by BLH to verified whether the reduced emission appears or not.

Response: We agree. CO is a tracer of primary emission, and the reduction in CO in the afternoon coincided with the increase in BLH. Correction of BLH is often performed by correction of $\Delta$CO since it has a relatively long lifetime than aerosol (Lin et al., 2020). We have performed $\Delta$CO correction.

It now reads, "…The decrease in the afternoon was likely due to the reduced primary emission and increased boundary layer height (BLH). The increasing BLH in the afternoon diluted the air pollutants, which was confirmed by the decreased CO at the same time (Fig. S16). The $\Delta$CO corrected PM$_{2.5}$ first showed an increase at noon and decreased afterward probably (Fig. S16) because the PM$_{2.5}$ emissions and/or secondary formation were insufficient to make up the diluting effects in the afternoon…"

[Figure]

Figure S16. Diurnals of CO (a); observed and $\Delta$CO corrected $PM_{2.5}$ for 2015-2018 (b) and 2019-2021 (c). $PM_{2.5}$ concentrations were normalized to the first hour of the day.

Line 191: (Sect. 21) ? or Sect 2.1
Response: Corrected.

Line 233-234: Figure 4 ?
Response: Corrected.

Line 242-244: More information of OA source apportionment by the PMF in this study (2021) should be provided to make the conclusion robust, referring to Feng T, et al. Atmos. Chem. Phys., 2023, 23: 611-636.
Response: We have now provided more information about OA source apportionment using PMF. And the study by Feng et al., (2023) is now cited.
In the revised text, it now reads, "…For AMS measurement, Elser et al. (2016) first deployed an AMS with a $PM_{2.5}$ inlet and comprehensively evaluated its performance during the winter campaign in Xi'an. Based on the same design, a soot particle AMS (SP-AMS) equipped with a $PM_{2.5}$ inlet was recently deployed at the same sampling site, quantifying $PM_{2.5}$ chemical composition with high mass and temporal resolution (Lin et al., 2021; Duan et al., 2022). OA factors were apportioned using the Positive Matrix Factorization (PMF) with the Multilinear Engine (ME-2) (Elser et al., 2016; Zhong et al., 2020; Duan et al., 2021; Duan et al., 2022), Similarly, OA source apportionment was performed for the 2021 dataset using the Igor Pro (WaveMetrics Inc.) - based interface of SoFi (version 8.2.1) (Canonaco et al., 2013). The apportioned factors

explained the input OA matrix well (Fig. S4-S6)…"

also "…The increase in secondary OA was consistently observed in other areas in China (Li et al., 2022; Nie et al., 2022), which may change of volatility and viscosity of OA (Feng et al., 2023)…"

[Figure]

**Figure S3.** Time series of the NR-PM₂.₅ species and the mean PM₂.₅ concentrations from January 14 to February 6, 2021.

[Figure]

**Figure S4.** (a) The value of Q/Qexp (Canonaco et al., 2013) as a function of the number of factors (nb. Of factors) with the circle highlighting the number of 5 factors that best represented the current dataset; and (b) scatter plot between the sum of all OA factors (5-factor solution) and the PMF input with the slope and determination of correlation ($r^2$) being close to unity.

[Figure]

**Figure S5.** Normalized mass spectra of the OA factors. MO-OOA and LO-OOA were summed up as one OOA in the main text for better comparison across different years. OM:OC, O:C, and H:C ratios are calculated using the improved-ambient (IA) method (Canagaratna et al., 2015).

[Figure]

**Figure S6.** Time series of the OA factors for the 5-factor solution.

Line 246: How do the authors explain the large reduction of COA? Dose it mean that the cooking frequency is lower than before or the cooking method change or other reasons?

Response: We have now discussed the reduction in COA (also see the response to Reviewer #1). Regarding the reduction in Cooking OA, it could be related to upgraded kitchen facilities, as well as the uncertainties in comparing the changes in COA. We have now discussed the uncertainties in the comparison. In the revised manuscript it now reads, "…The 5 datasets obtained at the same sampling site were averaged to gain insights into the changes in chemical composition and OA factors over these years, although it is noted that measurements were not conducted at the same period in each year with the same duration. The one standard deviation (sd) (Table S4) ranged from 1.8 to 78.5 µg m$^{-3}$, or 13-110% for the NR-PM species and from 0.7 to 29.4 µg m$^{-3}$, or 14-88% for the OA factors. …" and "…The reduction in fossil fuel and biomass burning was consistent with the reduction in CO and SO$_2$ as discussed above. Similarly, the primary cooking OA factor was also largely reduced by 84% (16.4 µg m$^{-3}$). The reduction in cooking emissions could be partly due to upgraded kitchen facilities (Liu et al., 2022), although we note that the sampling time and duration of the compiled studies were not the same (Table S3), causing uncertainties in the evaluations of changes in OA factors years. Future studies with long-term continuous measurements e.g., using ACSM (Chen et al., 2022), will improve the understanding of trends in OA factors in this region…"

Line 290-291: The self-reaction between peroxy radicals can produce SOA, but the reaction between NOx/NO3 and peroxy radicals has the same effect. The branch ration between these two pathways can influence the level of increasing or decreasing SOA formation. More explanation could add here.

Response: We agree that the reaction between NOx/NO$_3$ and peroxy radicals can also produce SOA. However, because NO$_x$ was largely reduced (by 70%) during the Covid period, this formation pathway is reduced compared to the self-reaction between peroxy radicals. We have added more explanation here. It now reads, "…In addition to the self-reaction, some of the reactions between NO$_x$/NO$_3$ and peroxy radicals can also produce organic nitrate that could partition to the particle phase (Lin et al., 2021). However, such reaction pathways were likely playing a less important role when NOx were largely reduced…"

Comments for Figures
Figure 2: Deep and light color were applied for figure a/c and b/d, but detailed legends should be provided.
Response: We have now revised the figure accordingly.

[Figure]

**Figure 2.** Observed and deweathered diurnal patterns for PM$_{2.5}$ (a,c) and O$_3$ (b, d) for January-February in 2015-2018 and 2019-2021

Figure S2: The x-axis labeling is ambiguous. 2015-2021?
Response: We have now labeled the Figures with more detail to avoid confusion.

[Figure]

**Figure S2.** Time series of PM$_{2.5}$ and O$_3$ at the 13 sampling sites from 2015 to 2021. The abbreviation for the sampling site is the same as in Figure S1. Data were averaged at 2 weeks for clarity reasons.

References:

Canagaratna, M. R., Jimenez, J. L., Kroll, J. H., Chen, Q., Kessler, S. H., Massoli, P., Hildebrandt Ruiz, L., Fortner, E., Williams, L. R., Wilson, K. R., Surratt, J. D., Donahue, N. M., Jayne, J. T., and Worsnop, D. R.: Elemental ratio measurements of organic compounds using aerosol mass spectrometry: characterization, improved calibration, and implications, Atmos. Chem. Phys., 15, 253-272, 10.5194/acp-15-253-2015, 2015.

[revised manuscript text omitted]